# CALIBRATE TO DISCRIMINATE:
# IMPROVE IN-CONTEXT LEARNING WITH LABEL-FREE COMPARATIVE INFERENCE

## ABSTRACT

While in-context learning with large language models (LLMs) has shown impressive performance, we have discovered a unique miscalibration behavior where both correct and incorrect predictions are assigned the same level of confidence. We refer to this phenomenon as *indiscriminate miscalibration*. We found that traditional calibration metrics, such as Expected Calibrated Errors (ECEs), are unable to capture this behavior effectively. To address this issue, we propose new metrics to measure the severity of indiscriminate miscalibration. Additionally, we develop a novel in-context comparative inference method to alleviate miscalibrations and improve classification performance. Through extensive experiments on five datasets, we demonstrate that our proposed method can achieve more accurate and calibrated predictions compared to regular zero-shot and few-shot prompting.

## 1 INTRODUCTION

LLMs have exhibited emergent capabilities such as advanced creative writing, summarization, translation, arithmetic and commonsense reasoning, etc (Wei et al., 2022; Brown et al., 2020). One of the most fascinating aspects of LLMs is their in-context learning capabilities. In particular, this involves adding pairs of demonstration examples to the prompt, and has been shown to significantly enhance the performance of LLMs (Brown et al., 2020). This capability offers users the significant advantage of utilizing LLMs without the need to train or fine-tune their own models. As a result, there is a growing need for research into calibration techniques (Zhou et al., 2023; Zhao et al., 2021; Jiang et al., 2023a; 2021) to ensure the reliability of model outputs as well and improve performance.

The concept of calibration in modern deep learning models was first introduced in (Guo et al., 2017) where the authors also proposed metrics (such as Expected Calibration Errors, reliability diagrams, etc) and methods (such as temperature scaling) for characterizing and mitigating miscalibration issues. The miscalibration of LLMs has also been studied in recent works (Xiong et al., 2023; Tian et al., 2023; Liusie et al., 2024; Zhao et al., 2023; Geng et al., 2023; Kamath et al., 2020). In this work, we show that LLMs with zero-shot and few-shot prompting exhibit a unique miscalibration issue on classification tasks, which we refer to as *indiscriminate miscalibration*. This phenomenon occurs when models assign equal confidence to correct and incorrect predictions. We show that this phenomenon cannot be quantitatively measured by Expected Calibration Errors (ECE). One alternative metric that can help catch this phenomenon is using Macro-average Calibration Error (MacroCE) proposed in Si et al. (2022). However, it may not depict the phenomenon thoroughly as it only computes the means of the distributions. We further propose a metric to help describe the phenomenon in more details. We hypothesize that this indiscriminate miscalibration occurs because the model treats each sample independently and has not been trained on the corresponding dataset. As a result, the predicted probabilities are not comparable across samples, which can further negatively impact prediction accuracy.

To this end, we propose a label-free in-context comparative inference method that adds unlabeled samples to the prompt. This encourages the model to adjust and calibrate its predictions without requiring labels for the demonstration examples. The added examples serve as a proxy to help the model better understand the test examples and the corresponding task. We delve deeper into the principles of our method and develop an aggregation step for improved calibration and a post-hoc

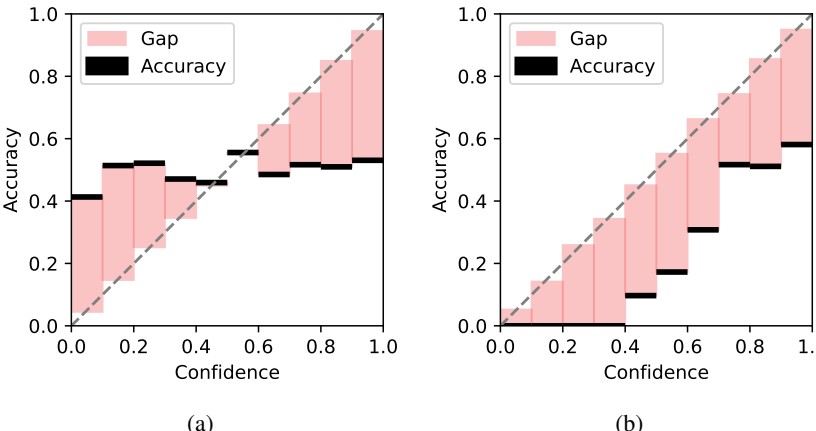

Figure 1: Simulated reliability diagrams show different miscalibration behaviors but having the same ECE and accuracy. (a) an indiscriminate miscalibration behavior which is also observed in zero-shot and few-shot prompting in our experiments; (b) a regular miscalibration behavior closer to the original calibration paper (Guo et al., 2017).

calibration method that can further enhance performance. Through experiments on multiple datasets, we demonstrate significant gains in performance measured by F1 scores, accuracy, and traditional calibration metrics. We also show that our label-free in-context comparative inference method helps to alleviate indiscriminate miscalibration, enabling models to assign higher confidence to correct predictions and lower confidence to incorrect ones.

## 2 RELATED WORK

### 2.1 CONFIDENCE AND CALIBRATION

Confidence calibration on modern neural networks has been discussed in (Guo et al., 2017) where the authors showed that large vision models are poorly calibrated. and proposed the Expected Calibration Errors (ECEs) that have been widely used in the literature. A more recent paper reviews some drawbacks of ECEs and proposed Instance-level Calibration Error (ICE) and Macro-average Calibration Error (MacroCE) (Si et al., 2022). Recent studies have shown great interests of model uncertainty and calibration in language models as well(Xiong et al., 2023; Tian et al., 2023; Liusie et al., 2024; Zhao et al., 2023; Geng et al., 2023; Kamath et al., 2020). As one of the most popular method for prompt engineering with LLM, in-context few-shot prompting (Brown et al., 2020) has the output instability issue, which was revealed by Zhao et al. (2021). They proposed a simple method to estimate and adjust majority label bias, recency bias, and common token biases introduced by in-context learning. Jiang et al. (2021) also showed that LLMs can be overconfident about their answers and calibration purposed fine-tuning or post-hoc calibration method can be used to improve the performance. Other similar methods such as batch calibration (Zhou et al., 2023) has been proposed as well. More recent studies show that few-shot prompting, finetuning or chain-of-thoughts can all suffer miscalibration issues (Zhang et al., 2023). In a low shot setting (e.g. $< 4$ demonstration examples), model prediction accuracy and calibration error can both increase and the trade-off can be improved with larger models or more shots (e.g. $> 8$ shots). Moreover, instruction fine-tuned LLMs exhibit the same miscalibration issue(Jiang et al., 2023a) as well.

### 2.2 IN-CONTEXT LEARNING

In few-shot prompting (Brown et al., 2020) setting, one provides input-label demonstration pairs in the prompt to improve LLMs' understanding and generation capability. Many methods have been proposed to improve ICL and understand its behaviors. Wang et al. (2023) shows that the label words and contextual words can impact model generations. Min et al. (2022) shows that providing a few examples of the label space, the distribution of the input text, or the format of the sequence

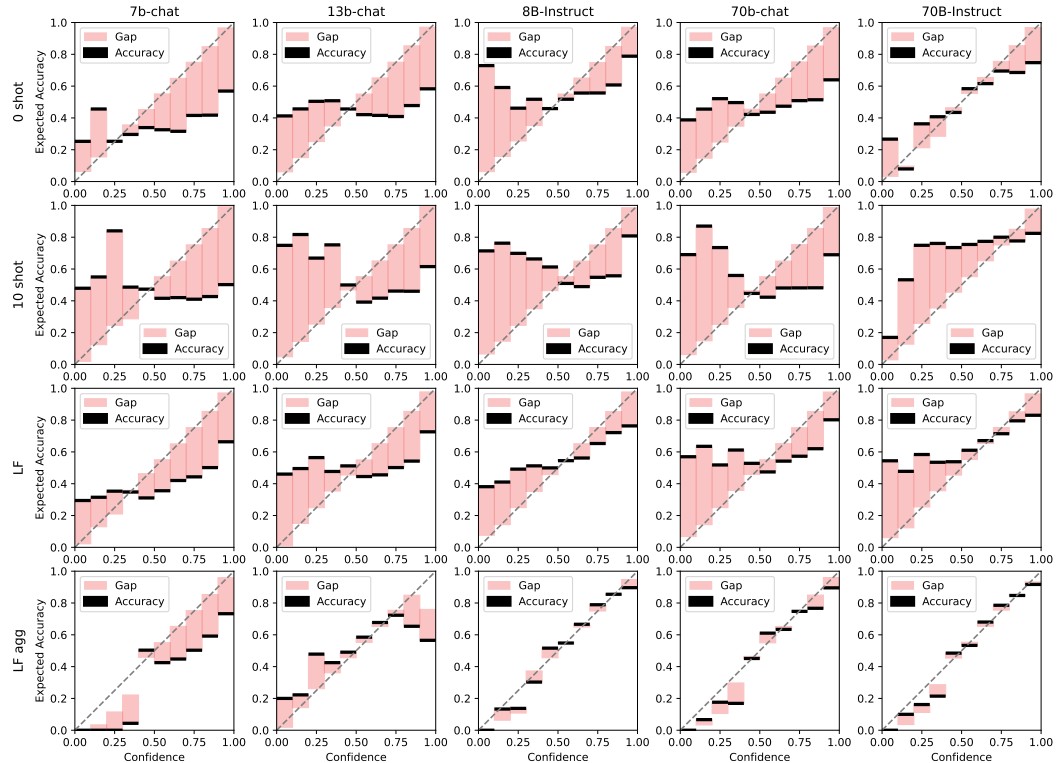

Figure 2: Reliability diagrams averaged across 5 datasets. The confidence matches the accuracy (y-axis) for a perfectly calibrated model. Hence, the red gaps indicate the severity of miscalibrations for each confidence bin. The first and second rows show the indiscriminate miscalibration behavior of large language models under in-context learning (0-shot and 10-shot) setting where accuracy are similar regardless whether confidences are high or low. In certain cases, lower confidences can give even higher accuracy. Under comparative inference setting (e.g.third row), such issue is alleviated and significantly improved with aggregated comparative inference (e.g.last row).

can be more important than ground truth labels. Ensemble methods such as self-consistency (Wang et al., 2022) over a diverse set of reasoning paths can improve performance as well. In a more recent study, comparison ability (Liusie et al., 2024) without labels was also demonstrated as one of the emergent capability of LLMs (Wei et al., 2022).

## 3 BACKGROUND

### 3.1 NOTATIONS

Denote a classifier as $f_\theta(X)$ which is parameterized by $\theta$ and takes input $X \in \mathcal{X}$. $f_\theta(X)$ is optimized on the training data $\mathcal{D}_{\text{train}} = \{x_1, y_1, ..., x_N, y_N\}$ with label $Y \in \mathcal{Y} = \{1, 2, ..., K\}$. We take a calibration point of view and assess whether the label probability generated by a language model can accurately express its confidence level such that the probability estimate $\hat{P}(Y|X)$ can reflect the actual probability of the prediction being correct (Guo et al., 2017; Jiang et al., 2021; Liusie et al., 2024). Here we use $\hat{P}$ to represents the probability estimate, which is a vector of probabilities over $K$ labels. The label estimate is derived by taking the argmax such that $\hat{Y} = \text{argmax}_Y \hat{P}(Y|X)$ (Guo et al., 2017; Liusie et al., 2024). For simplicity, we further define $p^* = \max_Y \hat{P}(Y|X) = \hat{P}(\hat{Y}|X)$ to represent the probability estimate of the predicted label which is used to represent the *confidence* (Guo et al., 2017) of the label estimate.

### 3.2 CONFIDENCE ESTIMATE FOR LLMS

For an LLM, we extract the token probability for each class $Y \in \mathcal{Y}\{1, 2, ..., K\}$ as the probability estimates $\hat{P}(Y|X)$. For example, in a binary classification setting, we have the label domain to be $\mathcal{Y} = \{0, 1\}$ where 1 indicates 'Yes', and 0 indicates 'No'. The probability estimate $\hat{P}(Y = 1|X)$ is extracted by computing the likelihood of generating label 'Yes'. More specifically, if the generated sentence is 'The answer is: Yes.' and the word 'Yes' is tokenized as 'Yes'. Then we use the probability of token 'Yes' after 'The answer is:'. In the case where a label word has multiple ways of being tokenized, we sum over all possible tokens. [1]

Now that we have extracted label probability estimate, we then compute the Expected Calibrated Error (ECE) (Guo et al., 2017), which has been widely used to to characterize the calibration level of a model's confidence (Geng et al., 2023). It is defined as the following,

$$\text{ECE} = \mathbb{E}_{\hat{p}}[\|\mathbb{P}(\hat{Y} = Y|\hat{P} = p) - p\|].$$

The lower the ECE, the more calibrated the model is. Similarly to many other observations (Xiong et al., 2023; Tian et al., 2023; Liusie et al., 2024; Zhao et al., 2023; Geng et al., 2023; Kamath et al., 2020), we found that the LLMs' probability estimates can be miscalibrated (Figure 2, Table 1). However, while ECE can be generally used to measure whether a model is calibrated, it fails to distinguish different miscalibration behaviors. For example in Figure 1, two reliability diagrams have same ECE but have clearly different miscalibration behaviors which we will elaborate more in the next section.

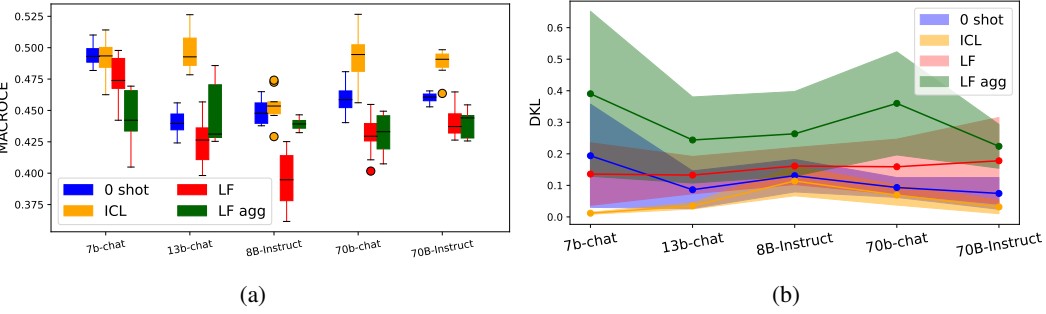

(a)          (b)

Figure 3: Quantifying indiscriminate calibration using Indiscriminate Ratio (MacroCE) and Discriminate KL (DKL) divergence. The metrics aim to capture the difference between the probability distributions of correct and incorrect predictions. Numbers are averaged across 5 datasets. Bar or Shaded area describes the standard deviations across datasets. A smaller MacroCE (or a larger DKL) indicates a more discriminate calibrated model. Comparative inference helps alleviate indiscriminate miscalibration and the aggregation method can further improve it.

### 3.3 INDISCRIMINATE MISCALIBRATION FOR IN-CONTEXT LEARNING

In contrast to other models' miscalibration behavior revealed in Guo et al. (2017), LLM's miscalibration can be special such that it's overconfident and the phenomenon has been reported in several previous studies (Jiang et al., 2021; Tian et al., 2023). In our experiments, we found that *regardless whether the label estimate is correct or not, LLM tends to give equal confidence (e.g.probability estimate) to its predictions when LM is not trained on the task (e.g. in zero-shot setting)*. The confidence is not necessarily high (e.g. overconfident). In certain scenarios, LM can also be underconfident (e.g. low confidence with high accuracy). More specifically, as shown in Figure 1 (a), the model gives same accuracy for all confidence levels, this is also discussed in Jiang et al. (2021). We report this phenomenon using several classification tasks as an example, shown in Figure 2.

We refer to such behavior as **indiscriminate miscalibration**, a special case of miscalibration , that can't be captured by ECEs. In conventional miscalibration scenario such as the simulated scenario

---

[1]Suppose the label word is 'Positive'. And 'Positive' can be tokenized as i) 'Pos' + 'itive' and ii) 'Positive'. We sum over both 'Positive' and 'Pos' tokens to better represent the probability estimate. As in this context, generating 'Pos' is almost 100% indicating the next token is 'itive'.

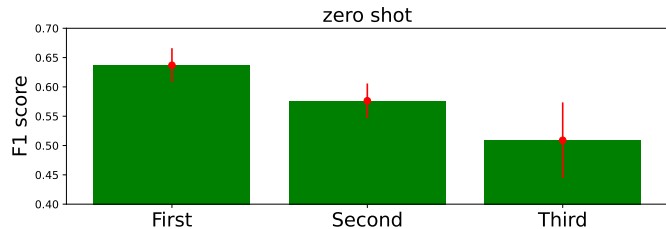

Figure 4: Inference performances vary significantly at different positions in the comparative inference setting, examples from the TREC task (Li & Roth, 2002; Hovy et al., 2001). In the zero-shot setting, the performance decays at the second and third positions.

in Figure 1 (b) or the ones reported in Guo et al. (2017), though in each confidence level bins, its actual accuracy has gaps towards the expected accuracy, the model generally has higher confidence in predictions with high accuracy and lower confidence in predictions with low accuracy. It still possess the ability to give different confidence levels toward possible correct predictions and incorrect predictions. Such ability is more strictly defined as *Relative Confidence* (Geng et al., 2023; Kamath et al., 2020; El-Yaniv et al., 2010; Wightman et al., 2023), the ability to rank samples, distinguishing correct predictions from incorrect ones. In Si et al. (2022), authors show that ECE is incapable for describing this behavior and they proposed Instance-level Calibration Error (ICE) as the following,

$$\text{ICE} = \frac{1}{n} \sum_{i=1}^{n} |\mathbb{I}(y_i = \hat{y}_i) - \hat{P}(\hat{y}_i | x_i)|.$$

Here, we use $\hat{P}(\hat{y}_i | x_i)$ to replace the confidence in the original paper. While ICE tries to differentiate correct and wrong predictions, miscalibration issue can be disguised when the accuracies are high. To address this, Macro-average Calibration Error (MacroCE) is further proposed to balance the correct predictions and incorrect ones. MacroCE is defined as,

$$\text{ICE}_{\text{pos}} = \frac{1}{n_p} \sum_{i=1}^{n_p} |\mathbb{I}(y_i = \hat{y}_i) - \hat{P}(\hat{y}_i | x_i)|, \forall \hat{y}_i = y_i,$$

$$\text{ICE}_{\text{neg}} = \frac{1}{n_n} \sum_{i=1}^{n_n} |\mathbb{I}(y_i = \hat{y}_i) - \hat{P}(\hat{y}_i | x_i)|, \forall \hat{y}_i \neq y_i,$$

$$\text{MacroCE} = \frac{1}{2}(\text{ICE}_{\text{pos}} + \text{ICE}_{\text{neg}})$$

where $\text{ICE}_{\text{pos}}$ is the ICE score computed over correct predictions and $\text{ICE}_{\text{neg}}$ is the one with incorrect predictions. On the other hand, MacroCE computes the mean of the difference between confidence and accuracy by averaging over samples. It may not capture certain different miscalibration distribution patterns. Imagine two cases (case A and and B), both cases have same means for correct prediction confidences and incorrect prediction confidences. The MacroCE would be similar for case A and B. However, when the variances are different, these cases should show different indiscriminate levels. We illustrate one visual example in the Appendix. To measure the difference between correct predictions and incorrect predictions on a more granular level such as using information of variances, we also define the Discriminate KL divergence (DKL) as the following,

$$\text{DKL} = \text{KL}[\mathbb{P}(p^* | \hat{Y} = Y) || \mathbb{P}(p^* | \hat{Y} \neq Y)]. \tag{1}$$

Unlike MacroCE, DKL measures the distribution differences. Larger DKL indicates more discriminate confidences levels between correct predictions and incorrect predictions. We report the results in Figure 3b. Similarly, we observed that our proposed method helps improving DKL, i.e. more discriminate confidence level.

## 4 METHOD

As shown above, when we use LLMs with zero-shot and few-shot prompting, they have trouble in differentiating confidences for correct and incorrect predictions. Our hypothesis is that each sample

is treated independently and LLMs are not able to compare or rank them without being trained on them even with few-shot prompting. Thus we propose a label-free method by asking LM to compare a sample of interest $X = x_i$ with other unlabeled samples jointly to adjust for more calibrated results.

## 4.1 COMPARATIVE INFERENCE

Considering three samples $x_i, x_j, x_k$, we are interested in performing a comparison to derive the labels probability estimate $\hat{P}(y_i, y_j, y_k | x_i, x_j, x_k; C)$, where $C$ represents a prompting instruction asking for a comparison. The following template is a prompt example,

```
(Task Definition) A news description topic can be one of the
following four types. ###Types: World, Sports, Business, Sci/Tech.

(Inputs Samples) For the following 3 news descriptions: ###News
Description 1: XXXX. ###News Description 2: XXXX.  ###News
Description 3: XXXX.

(Comparison prompt) By comparing them, we know the most suitable
types for each of these 3 news descriptions, respectively, are:
```

Note that even though we use words like 'comparing', the goal is not to ask LLMs to say if sample 1 is 'more positive' than sample 2 or sample 3 as that may change the original task meaning. It's more important to present multiple unlabeled samples in the context and ask LLMs to predict labels for each of them jointly.

### 4.1.1 ASYMMETRIC PROBABILITIES

The first thing to notice is that, due to the auto-regressive nature of large language models, we have $\hat{P}(y_i, y_j | x_i, x_j; C) \neq \hat{P}(y_j, y_i | x_j, x_i; C)$. More specifically, if $x_i$ appears before $x_j$ in the prompt, one would expect $\hat{y}_i$ to be generated before $\hat{y}_j$, which means the $\hat{y}_j$ is generated conditioned on $\hat{y}_i$. And it can be shown with the following,

$$\hat{P}(y_i, y_j | x_i, x_j; C) = \hat{P}(y_i | x_i, x_j; C)\hat{P}(y_j | x_i, x_j, \hat{y}_i; C). \tag{2}$$

This could lead to bias toward samples generated in later orders. Indeed, ordering of words in the prompts can impact performance significantly (Lu et al., 2021; Min et al., 2022). In our experiments, we observed a degradation in inference performance for samples appearing later in the input (Figure 4). Besides, the post-processing and probabilities extractions of inputs after the first prediction can be tricky. Hence, we focus on the prediction of the first input throughout the paper, i.e. $\hat{P}(y_i | x_i, x_j, ...; C)$ and always put the sample of interest as the first sample to predict. The rest input samples only serve as a reference comparison inputs. Listing the input of interest at the beginning also have a benefit such that if users are only interested in the input label, they don't need to ask LLMs to generate all rest tokens which can be more efficient.

### 4.1.2 COMPARISON BIAS

In an ideal setting, we can derive the probability estimates leveraging all the possible input samples $\hat{P}(y_i | x_i, \mathcal{D}_x; C)$. However, this is impossible as that will make the prompt too long to be processed by LLMs. Thus, we make the following assumptions,

$$\hat{P}(y_i | x_i, \mathcal{D}_x; C) \approx \hat{P}(y_i | x_i, \{x\}_j; C) + \epsilon_j, \text{ with } \mathbb{E}(\epsilon_j) = \mathbf{0}, \tag{3}$$

where $\{x\}_j$ is a sequence of samples from $\mathcal{D}_x$; $\epsilon_j$ is the bias introduced when comparing with $\{x\}_j$ in the prompt instead of $\mathcal{D}_x$ (e.g. similar to the contextual bias (Zhao et al., 2021). We further hypothesize the biases introduced by different comparison samples can be averaged out (e.g

expectation is zero) so we can approximate it in practice by the following,

$$\widehat{P}(y_i|x_i, \mathcal{D}_x; C) \approx \int_{\mathbb{P}_j} \hat{P}(y_i|x_i, \{x\}_j; C) \approx \sum_{j=1}^{J} \frac{1}{J} \hat{P}(y_i|x_i, \{x\}_j; C), \tag{4}$$

where $J$ is a finite number of sets of comparisons. This is simply aggregating multiple comparative inference probabilities with different comparison samples. We expect with aggregations, calibration error will go down drastically. As one could see in Figure 5a, we can see ECE drastically decreased with aggregation. F1 scores and accuracies are also improved but relatively less aggressive. Though there still exist other biases (e.g. contextual, label, etc) that can be further calibrated. However, this indeed will increase inference cost by the number of aggregation times.

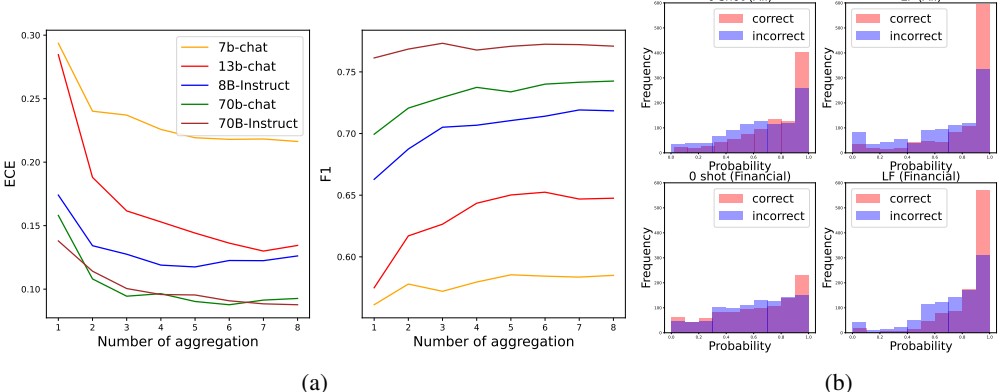

(a)          (b)

Figure 5: (a).Inference performances vary significantly at different positions in the comparative inference setting, examples from the TREC task (Li & Roth, 2002; Hovy et al., 2001). In the zero-shot setting, the performance decays at the second and third positions. (b).Comparison of confidence for correct and incorrect predictions. For the Financial task (Bottom row) where labels are more contrast and comparable, the differences between our method (Right column) and baseline (Left column) are more ousstanding compared with other tasks (Top row).

### 4.2 POST-HOC CALIBRATION

There exist many ways in the literature for calibrating probabilities such as Histogram binning (Zadrozny & Elkan, 2001), Isotonic regression(Zadrozny & Elkan, 2002), Platt scaling (Platt et al., 1999), etc. One simplest approach is temperature scaling (Guo et al., 2017) where we apply weighted softmax on logits or probabilities to recalibrate probabilities. While temperature scaling can decrease ECE, it does not help with improving model's classification performance (Guo et al., 2017). Extensions of Platt scaling towards multi-class classification is referred as vector scaling and matrix scaling, where one would optimize the following linear layers on top of the probability estimates,

$$\tilde{P}(y_i|x_i) = \text{softmax}(W\hat{P}(y_i|x_i) + b),$$

where $\tilde{P}(y_i|x_i)$ is the calibrated probabilities. In the vector scaling case, $W$ is restricted to a diagonal matrix. And $W$ and $b$ are optimized using a validation dataset. Inspired by these methods, we propose to further calibrate our probability estimates by applying such affine transformations but on top of different comparison references. More specifically, we retrieve the recalibrated probabilities by,

$$\tilde{P}(y_i|x_i, \mathcal{D}_j; C) = \text{softmax}\{\sum_{j=1}^{J}\{W_j\hat{P}(y_i|x_i, \{x\}_j; C)) + b_j\}, \tag{5}$$

where $W_j, b_j$ can be estimated via $\text{argmax}\{-y_k\text{log}P^*(y_k|x_k, \{x\}_j; C))\}$ with a set of $\{x_k, y_k\}$. This is similar to the principle of contextual calibration used in Guo et al. (2017); Zhao et al. (2021). The difference is that previous studies focused calibrating biases introduced from labels, or other prompt contextual information whereas we are focusing on the comparison samples. Note that both

of our post-hoc approach in this section and other approaches mentioned earlier and the ones in the literature such as Abbas et al. (2024); Han et al. (2022); Jiang et al. (2023b); Shen et al. (2024) can be applied on top of the probabilities outputted by the method in Section 4.1.

## 5 RESULTS

### 5.1 EXPERIMENTS SETUP

**Models and Datasets**  We conducted experiments using the Llama(Touvron et al., 2023) family models: Llama2-7b-chat, Llama2-13b-chat, Llama2-70b-chat , Llama3-8b-instruct, Llama3-70b-instruct. Our experiments focused on classification problems. We picked 5 classic tasks covering different domains: AGNews (topic classification) (Zhang et al., 2015), trec (topic classifications) (Voorhees & Tice, 2000; Li & Roth, 2002; Hovy et al., 2001), Financial datasets (sentiment analysis) (Malo et al., 2014), emotion dataset (emotion detection) (Saravia et al., 2018) and hateful speech detection (Toxicity detection) (de Gibert et al., 2018). Due to limited capacity, we capped the number of testing samples at 500. For each method, we run 10 replicates with different seeds.

|  | Model | ICL | | | LF-ICL | | | |
|---|---|---|---|---|---|---|---|---|
|  |  | 0 shot | 3 shot | 10 shot | 0 shot | 0 shot-agg | 3 shot | 10 shot |
| F1↑ | L2-7B | 0.37 | 0.54 | 0.46 | 0.56 | **0.59** | 0.375 | 0.45 |
|  | L2-13B | 0.48 | 0.61 | 0.58 | 0.58 | **0.65** | 0.53 | 0.61 |
|  | L2-70B | 0.60 | 0.72 | 0.74 | 0.66 | 0.72 | 0.71 | **0.75** |
|  | L3-8B | 0.52 | 0.63 | 0.63 | 0.70 | 0.74 | 0.71 | **0.77** |
|  | L3-70B | 0.71 | 0.78 | 0.80 | 0.76 | 0.77 | 0.80 | **0.81** |
| ECE↓ | L2-7B | 0.35 | 0.36 | 0.46 | 0.29 | **0.22** | 0.45 | 0.39 |
|  | L2-13B | 0.35 | 0.32 | 0.36 | 0.29 | **0.13** | 0.30 | 0.24 |
|  | L2-70B | 0.16 | 0.21 | 0.20 | 0.17 | **0.13** | 0.21 | 0.19 |
|  | L3-8B | 0.29 | 0.29 | 0.30 | 0.16 | **0.09** | 0.21 | 0.16 |
|  | L3-70B | 0.17 | 0.15 | 0.19 | 0.14 | **0.09** | 0.17 | 0.16 |

Table 1: F1 scores and ECE averaged across 5 datasets with 10 runs for all models. LF indicates label-free comparative inference; 'agg' indicates aggregate 10 different comparative inference results. Note that for all LF-ICL experiments, we don't provide labels. Higher F1 score and lower ECE are achived by LF-ICL.

**Methodology**  Our experimental results are mainly composed with two parts: i) methods for manipulating prompts and ii) further post-hoc calibrations where we train an additional linear layer with less than hundreds of parameters. For i), the baseline is the independent inference where each prompt contains only one input sample without demonstrations (zero-shot). We provide our sample prompts in the Appendix. We include few-shot prompting (Brown et al., 2020) methods with either 3-shot or 10-shot. For our Label-Free (denoted as LF in figures and tables) comparative inference method, we use 2 additional input samples randomly picked from the training dataset as comparison references. As discussed in section 4.1, we always placed the sample of interest at the first place among all 3 samples and focused on the results of the first answer generated by LLMs. Our method also uses aggregations where we used different comparison references in the prompt and retrieved averaged results from up to 10 comparative inference results. LF method can be used with few-shot together, which is also included. For ii) post-hoc calibration methods, we mainly considered the vector scaling and matrix scaling methods that have been widely used in Guo et al. (2017); Zhao et al. (2021); Zhou et al. (2023); Zhang et al. (2023).

### 5.2 LABEL-FREE COMPARATIVE INFERENCE IMPROVES CLASSIFICATION PERFORMANCE

For classification performance metrics, we consider accuracy and F1 scores. We empirically found these two are consistent. As most of the datasets are imbalanced, we reported F1 scores averaged across 5 datasets in the main text and the corresponding accuracy results in the Appendix.
From F1 scores in Table 1 and simiarly accuracy from the Appendix, comparative inference

| | Model | Matrix Scaling | | | | | Vector Scaling | | | | |
|---|---|---|---|---|---|---|---|---|---|---|---|
| | | 0 | 3 | 10 | 0-LF | 10-LF | 0 | 3 | 10 | 0-LF | 10-LF |
| F1↑ | L2-7B | 0.57 | 0.60 | 0.50 | 0.69 | 0.69 | 0.54 | 0.53 | 0.47 | 0.67 | **0.70** |
| | L2-13B | 0.54 | 0.67 | 0.57 | 0.71 | 0.75 | 0.52 | 0.65 | 0.57 | 0.73 | **0.76** |
| | L2-70B | 0.54 | 0.74 | 0.70 | 0.77 | 0.79 | 0.51 | 0.68 | 0.71 | 0.77 | **0.80** |
| | L3-8B | 0.63 | 0.74 | 0.74 | 0.71 | **0.79** | 0.64 | 0.74 | 0.74 | 0.71 | 0.78 |
| | L3-70B | 0.74 | 0.79 | 0.80 | 0.79 | **0.83** | 0.70 | 0.78 | 0.80 | 0.79 | 0.82 |
| ECE↓ | L2-7B | **0.17** | 0.21 | 0.29 | 0.23 | 0.22 | **0.17** | 0.20 | 0.29 | 0.23 | 0.19 |
| | L2-13B | 0.24 | 0.20 | 0.29 | 0.25 | 0.20 | 0.23 | 0.23 | 0.28 | 0.20 | **0.18** |
| | L2-70B | 0.28 | 0.20 | 0.20 | 0.19 | 0.18 | 0.28 | **0.17** | 0.21 | 0.18 | **0.17** |
| | L3-8B | 0.20 | 0.17 | 0.18 | 0.24 | 0.19 | **0.16** | 0.18 | 0.18 | 0.21 | 0.18 |
| | L3-70B | 0.17 | 0.16 | 0.16 | 0.18 | 0.16 | 0.17 | **0.15** | **0.15** | 0.17 | 0.16 |

Table 2: F1 and ECE scores averaged across 5 datasets with 10 replicates for all models with post-hoc calibration. We experiment with different shot number: 0, 3, 10. LF represents label-free in-context learning. All inference results indicates are based on post-hoc calibration with 10 inference results.

without labels can outperform few-shot prompting (e.g. Llam2-7b and Llama2-70b), while both are significantly better than the zero-shot. We found that both Llama3-8b and Llama3-70b behave even better in comparative inference setting with the TREC and Emotion tasks. Interestingly, these two tasks have 6 different labels which are the most across 5 dataset. However, in the ICL few-shot setting, both models did not perform well in terms of the calibrations.

### 5.3 Alleviate Indiscriminate Miscalibration with Comparative Inference

By applying comparative inference, we observe a significant alleviation of the indiscriminate miscalibration issue. More specifically:i) Expected Calibration Error (ECE) results are generally improved, with the aggregation method (section 4.1.2) consistently performing the best (Table 1). ii). The MacroCE score decreased (Figure 3a) and KL divergence increased with comparative inference across models and datasets (Figure 3b). iii). These behaviors are represented in the reliability diagrams (Figure 2) as well. Additionally, we found that few-shot prompting can potentially worsen ECEs, depending on the specific models used(Table 1, Figure 2).

As we discussed in section 4.1.2, we expect aggregation to improve calibration issues more compared to classification performances given our assumption. And we do observe such behaviors as shown by Figure 2, Figure 3 and also by ECEs in Table 1. Aggregation can also improve F1 and accuracy as this can be seen as a way of ensemble, similarly to other methods such as Wang et al. (2022). In our experiment, the performance is even better than comparative + ICL for Llama2-7b and Llama2-13b. We can also expect combining ICL, aggregation and our methods can further enhance performances (Table 1). However, this will require more tailored labels, and inference costs.

### 5.4 LLMs Can Compare And Contrast Samples for Adjusting Confidences

Previous sections have shown that adding unlabeled input samples in the prompt and asking LLMs to compare them can lead to improved accuracy and calibration. We hypothesize that this is because, in the baseline (e.g., zero-shot) approach, LLMs struggle to differentiate between correct and incorrect predictions due to treating each sample independently during inference. By providing the comparison samples, it enforces model to compare and contrast across multiple variations, which 'regularizes' and calibrates its confidences. To better verify this, we compare the 0-shot baseline with 0-shot-LF as an ablation experiment and plot the histogram of bootstrapped [2] probabilities of correct prediction and incorrect predictions in Figure 5b. The results show that our comparison-based approach yields more pronounced differences.

---

[2]We bootstrapped probabilities from correct prediction and incorrect distribution with the same amount to better visualize the histogram as these two categories can be imbalanced.

Moreover, our findings suggest that the model is indeed comparing and contrasting across different samples. This is particularly evident when looking into the financial dataset, where the contrast result is more pronounced. We attribute this to the fact that the financial task is an ordinal task with 'positive' and 'negative' labels, where contrast and comparison naturally becomes more effective (in the sense that it amplifies the differences between prediction probabilities but doesn't necessarily lead to higher accuracy). With this setup, our approach amplifies the probability differences between correct and incorrect predictions, leading to more outstanding differentiation.

### 5.5 CONSIDERATIONS AND BEST PRACTICES FOR EFFECTIVE COMPARATIVE INFERENCE

First observation is that we can combine LF method with few-shot to further improve performances. We found performances enhanced across the board except for the Llama2-7b based on averged F1 score (Table 1). We hypothesize this is because Llama2-7b is not capable to leverage both comparative inference and ICL at the same time.

Secondly, we found that LF comparative inference with ICL are mostly effective with 10-shots setting whereas the independent method perform well with 3-shot for smaller models but better with 10-shot for larger or more capable models (e.g. Llama3-8b or 70b models). For examples, 10-shot comparative inference setting is significantly better than other baselines for Llama2-7b and Llama2-13b models on the TREC dataset.

### 5.6 POST-HOC CALIBRATION RESULTS

As described in section 4.2, vector scaling has $K$ parameters while our post-hoc method will have $J \times K$ parameters. In our experiments, $K$ is generally $< 10$ and we restrict $J = 10$, which will restricted the total number of trainable parameters to less than hundreds for vector scaling and thousands for matrix scaling. In addition, the optimizations of $W$ and $b$ are conducted using 200 validation samples for all datasets. Similarly, we reported F1 scores and ECE scores averaged across 5 datasets for each model in Table 2. We show the breakdowns in the Appendix.

We found that comparative inference with post-hoc calibration is consistently better than baseline post-hoc calibration methods or the baselines with few-shot prompting for Llama2 family in terms of F1 scores. For Llama3, baseline with few-shot prompting is better, which is likely due to the fact that Llama3 models have strong capability on few-shot prompting. We further show that comparative inference with post-hoc calibration can be combined with few-shot prompting and this will further boost the performance and is consistently better than all other methods in terms of F1 scores. ECEs on average are improved compared to baselines but for many settings are not better than certain methods in the previous setction. As indeed, post-hoc calibration has been recently used to improve classification performances (Zhao et al., 2021; Zhou et al., 2023) instead of focusing on conventional calibration issues.

## 6 DISCUSSION

In this paper, we study a special miscalibration behavior of large language models when being used for zero-shot and few-shot prompting, which we refer to as indiscriminate miscalibration. We propose metrics to quantify the severity of this issue and develop a label-free in-context comparative inference method to mitigate it. We show our label-free method can achieve better classification performance as well as more calibrated predictions on multiple datasets.

## 7 LIMITATIONS

The primary limitation is that this study only considered classification tasks. Secondly, we considered 5 models but are all from Llama families. While these 5 models are widely recognized for their performance in natural language processing tasks, the inclusion may introduce biases inherent to this particular model family. Future research can benefit from exploring a broader range of tasks and considering models from other families to provide a more comprehensive understanding of their capabilities and limitations. Finally, we did not explore different strategies for selecting comparison samples or utilize fine-tuned models, which could be a promising direction for future research.

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
