# Appendix of "Improving Performance and Calibration with Label-Free In-Context Comparative Inference"

## 1 Prompt Examples

---

**Zero-shot Prompt**

A question can be one of the following six types. ###Types: Abbreviation, Entity, Description and abstract concept, Human being, Location, Numeric value. For the given question, ###Question: For the given question, ###Question: What is an atom ? The most suitable type for the given question is:

---

**Three-shot Prompt**

A question can be one of the following six types. ###Types: Abbreviation, Entity, Description and abstract concept, Human being, Location, Numeric value. For instance, for the following example questions###Example Question: 1: What is the name of the managing director of Apricot Computer ?. ###Example Question: 2: When did Muhammad live ?. ###Example Question: 3: How many people lived in Nebraska in the mid 1900s ?. The most suitable types should be:1). Human being. 2). Numeric value. 3). Numeric value. For the given question, ###Question: Why does the moon turn orange ? The most suitable type for the given question is:

---

**Zero-shot Comparative Prompt**

A question can be one of the following six types. ###Types: Abbreviation, Entity, Description and abstract concept, Human being, Location, Numeric value.For the following 3 questions: ###Question 1:How far is it from Denver to Aspen ? ###Question 2:Where is the Kentucky Horse Park ? ###Question 3:What is the first personal computer company ? By comparing them, we know the most suitable types for each of these 3 questions, respectively, are:

---

**Three-shot Comparative Prompt**

A question can be one of the following six types. ###Types: Abbreviation, Entity, Description and abstract concept, Human being, Location, Numeric value. For instance, for the following example questions###Example Question 1: What is the name of the managing director of Apricot Computer ?. ###Example Question 2: When did Muhammad live ?. ###Example Question 3: How many people lived in Nebraska in the mid 1900s ?. The most suitable types should be:1). Human being. 2). Numeric value. 3). Numeric value. For the following 3 questions: ###Question 1:What county is Modesto , California in ? ###Question 2:Where is the Kentucky Horse Park ? ###Question 3:What is the first personal computer company ? By comparing them, we know the most suitable types for each of these 3 questions, respectively, are:

---

## 2 Accuracy Results

| | | ICL | | | LF-ICL | | | |
|---|---|---|---|---|---|---|---|---|
| | Model | 0 shot | 3 shot | 10 shot | 0 shot | 0 shot-agg | 3 shot | 10 shot |
| | L2-7B | 0.41 | 0.57 | 0.49 | 0.57 | 0.59 | 0.46 | 0.51 |
| | L2-13B | 0.49 | 0.61 | 0.59 | 0.59 | 0.64 | 0.57 | 0.64 |
| Accuracy↑ | L2-70B | 0.52 | 0.63 | 0.63 | 0.71 | 0.75 | 0.71 | 0.77 |
| | L3-8B | 0.61 | 0.72 | 0.74 | 0.67 | 0.73 | 0.71 | 0.75 |
| | L3-70B | 0.72 | 0.77 | 0.8 | 0.77 | 0.78 | 0.8 | 0.81 |

Table 1: Accuracy averaged across 5 dataset with 10 replicates for all models. 0-shot indicates baseline independent inference and the corresponding first row includes In-Context Learning with independent inference results; 'LF' indicates label-free comparative inference 0-shot setting and the corresponding second row includes the In-Context Learning with comparative inference results; 'agg' indicates aggregate 10 different comparative inference results.

| | | Matrix Scaling | | | | | Vector Scaling | | | | |
|---|---|---|---|---|---|---|---|---|---|---|---|
| | Model | 0 shot | 3 shot | 10 shot | 0 shot-LF | 10 shot-LF | 0 shot | 3 shot | 10 shot | 0 shot-LF | 10 shot-LF |
| | L2-7B | 0.6 | 0.64 | 0.55 | 0.7 | 0.71 | 0.57 | 0.6 | 0.53 | 0.69 | 0.71 |
| | L2-13B | 0.56 | 0.69 | 0.63 | 0.72 | 0.76 | 0.54 | 0.68 | 0.62 | 0.74 | 0.77 |
| Accuracy↑ | L2-70B | 0.55 | 0.76 | 0.72 | 0.78 | 0.8 | 0.53 | 0.71 | 0.71 | 0.78 | 0.8 |
| | L3-8B | 0.65 | 0.75 | 0.75 | 0.72 | 0.79 | 0.66 | 0.75 | 0.74 | 0.73 | 0.79 |
| | L3-70B | 0.76 | 0.8 | 0.81 | 0.8 | 0.83 | 0.74 | 0.79 | 0.81 | 0.8 | 0.83 |

Table 2: Accuracy averaged across 5 datasets with 10 replicates for all models with post-hoc calibration. n-shot: baseline in-context learning; LFrepresents label-free in-context learning. All inference results indicates are based on post-hoc calibration with 10 inference results.

## 3 Special Scenario with Indiscriminative Miscalibration

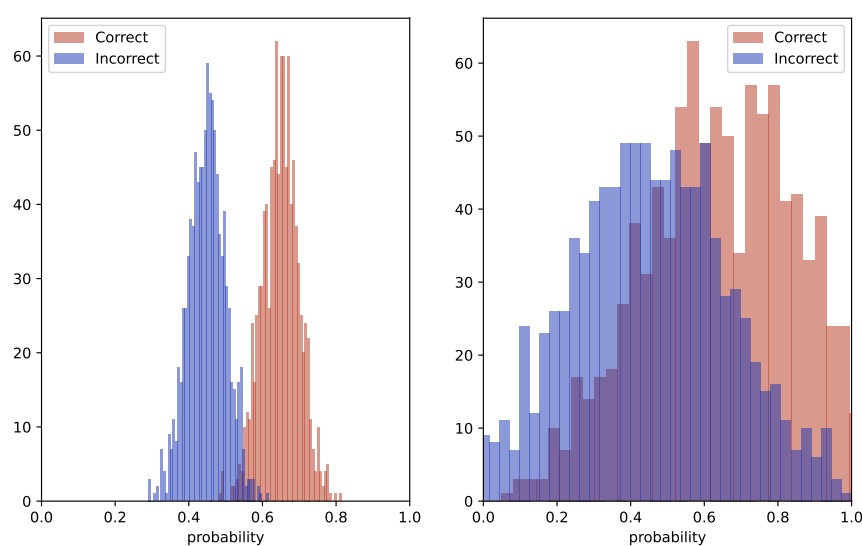

We simulate a scenario where correct prediction probabilities are sampled from a distribution with mean of $0.65$ and incorrect prediction probabilities are sampled from a distribution with mean of $0.45$. In the left figure, distributions have smaller variances while in the right figure, variances are larger. In this scenario, MacroCE will give the same value for both cases while 'qualitatively', left side is more 'discriminative' and has larged DKL value.

## 4 Individual Dataset Results

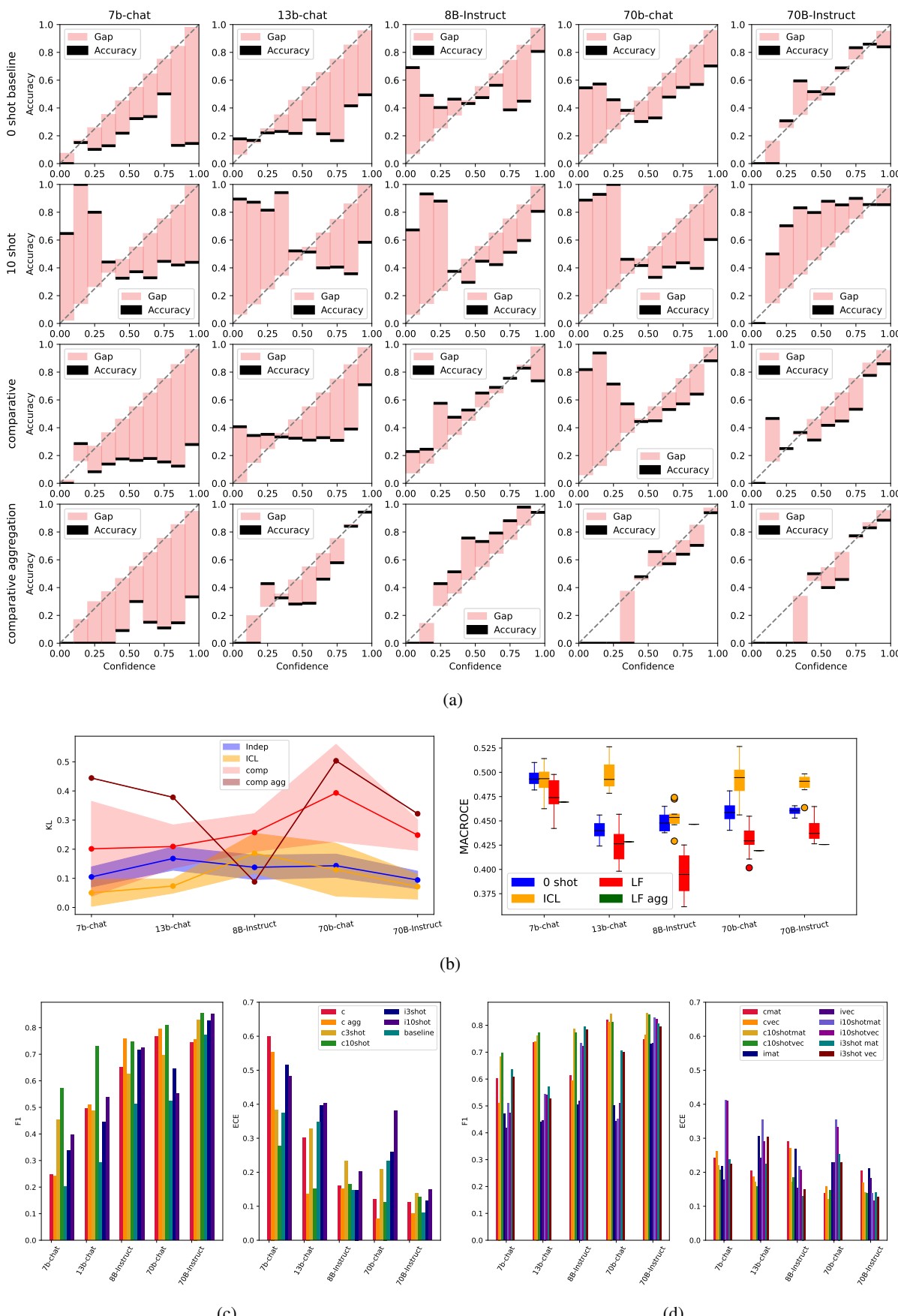

Figure 1: Trec Dataset Results. (a). Reliability Diagram. (b). DKL Divergence and MACROCE. (c). Comparative results with F1 scores and ECEs. (d). Post-hoc calibrated results. *Notice that for the aggregation method of the individual data result, we only have one round experiment so we only reported means without variances.

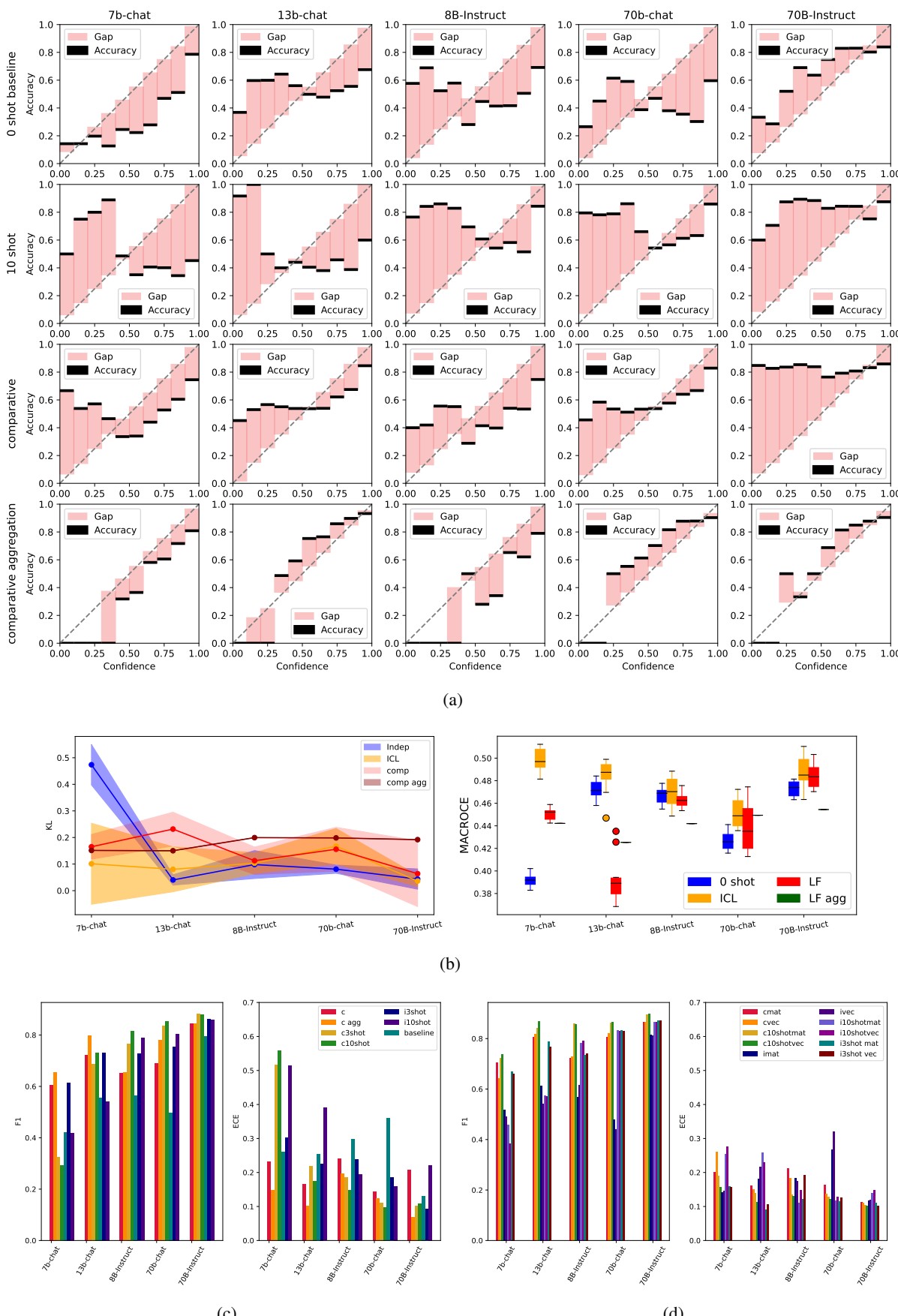

Figure 2: AGnews Dataset Results. (a). Reliability Diagram. (b). DKL Divergence and MACROCE. (c). Comparative results with F1 scores and ECEs. (d). Post-hoc calibrated results. *Notice that for the aggregation method of the individual data result, we only have one round experiment so we only reported means without variances.

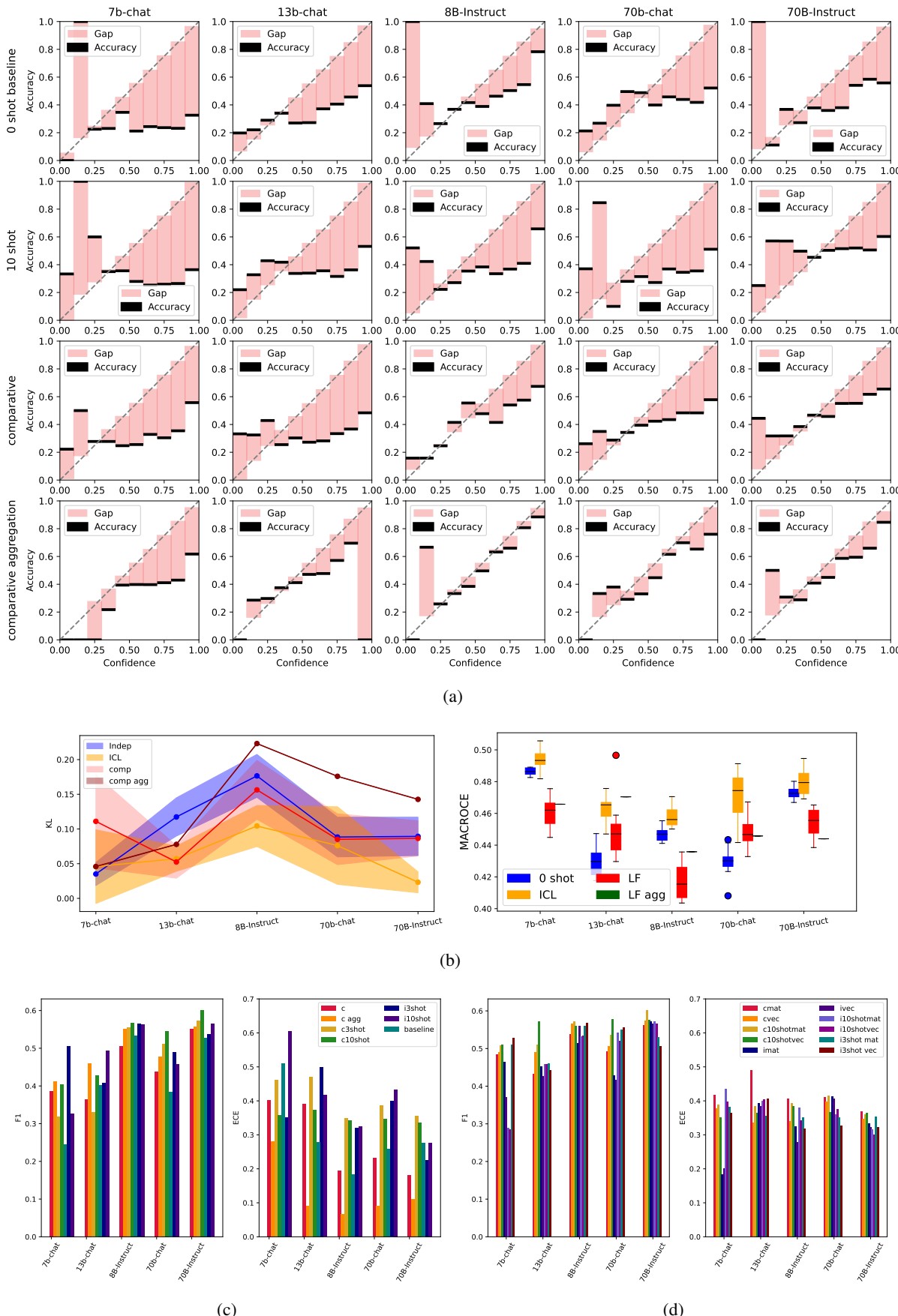

Figure 3: Emotion Dataset Results. (a). Reliability Diagram. (b). DKL Divergence and MACROCE. (c). Comparative results with F1 scores and ECEs. (d). Post-hoc calibrated results. *Notice that for the aggregation method of the individual data result, we only have one round experiment so we only reported means without variances.

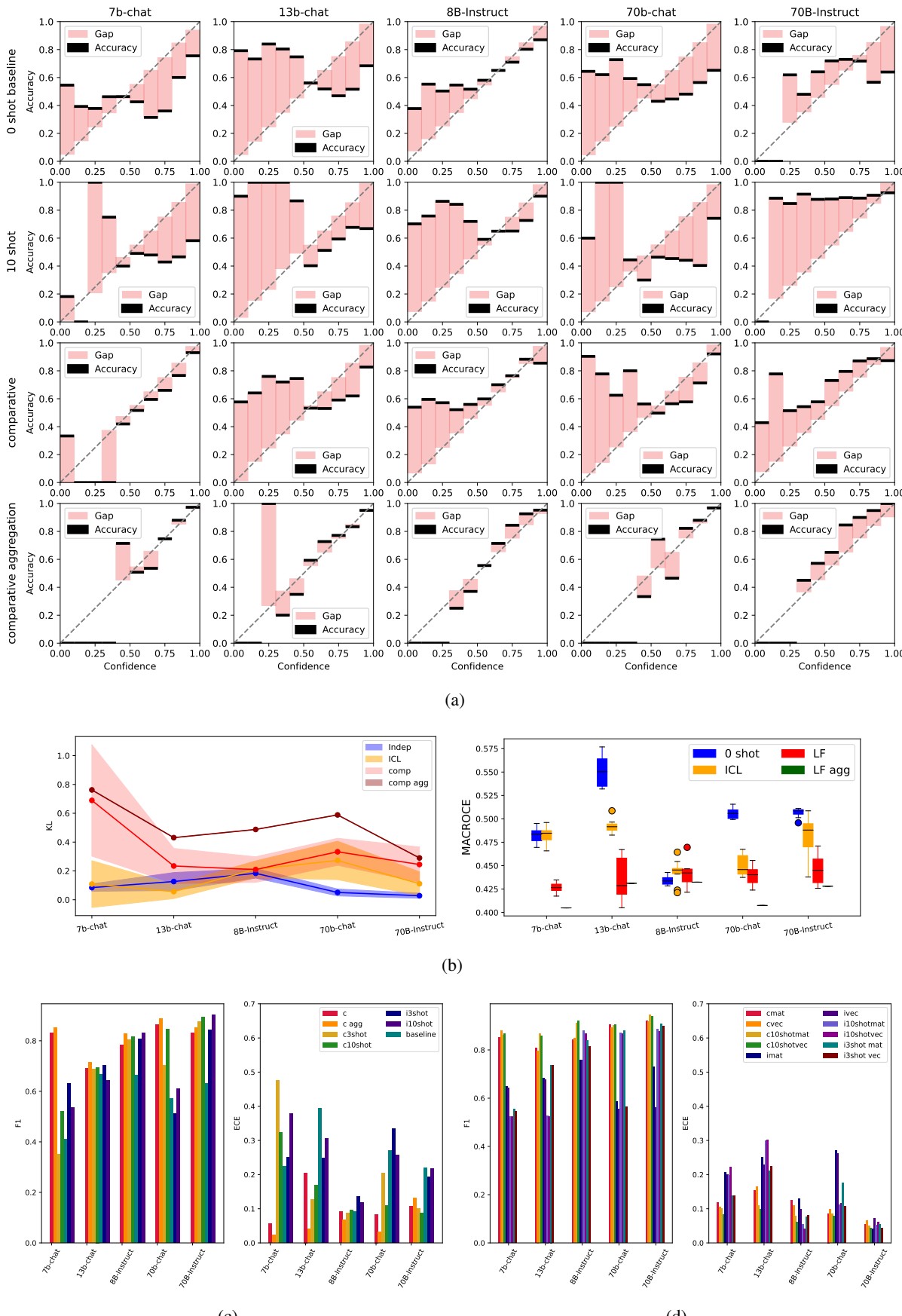

Figure 4: Financial Dataset Results. (a). Reliability Diagram. (b). DKL Divergence and MACROCE. (c). Comparative results with F1 scores and ECEs. (d). Post-hoc calibrated results. *Notice that for the aggregation method of the individual data result, we only have one round experiment so we only reported means without variances.

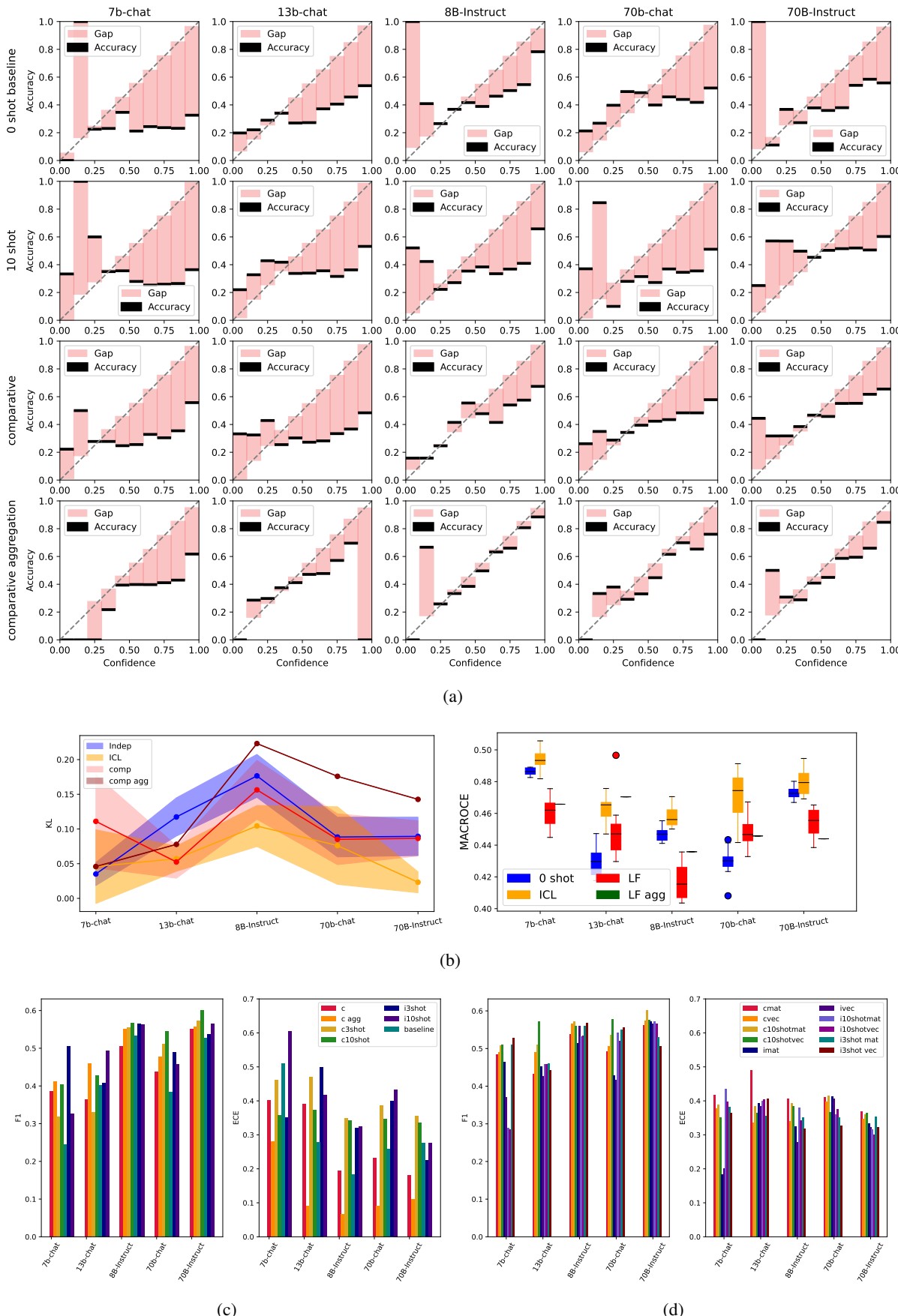

Figure 5: Emotion Dataset Results. (a). Reliability Diagram. (b). DKL Divergence and MACROCE. (c). Comparative results with F1 scores and ECEs. (d). Post-hoc calibrated results. *Notice that for the aggregation method of the individual data result, we only have one round experiment so we only reported means without variances.