# OpenReview forum: "Calibrate to Discriminate:Improve In-Context Learning with Label-Free Comparative Inference"
_ICLR.cc/2025/Conference — Submitted to ICLR 2025_

### Official Review · Reviewer_H97s · 2024-10-25

**Soundness:** 2
**Presentation:** 3
**Contribution:** 2
**Rating:** 3
**Confidence:** 4

**Summary:**

The paper studies the confidence calibration of large language models (LLMs) on classification tasks. The authors observe the indiscriminate miscalibration behavior of LLMs where accuracy is similar regardless of whether confidences are high or low and that metrics like ECE fail to capture this behavior accurately. The authors propose a KL divergence-based metric that can better capture this behavior and propose a post-hoc label-free calibration method to alleviate indiscriminate miscalibration.

**Strengths:**

The indiscriminate miscalibration observation, where both correct and incorrect predictions are assigned the same level of confidence, is quite interesting, as previous studies predominantly reported findings similar to Figure 1(b). Notably, the proposed method does not require labels, which is commendable. Experiments were conducted on five different LLMs of varying sizes which is also laudable.

**Weaknesses:**

**Key Related Works Missing**: One of my primary concerns is that the paper does not cite several key works in the calibration literature that are highly relevant, including:

1)	Abbas, M., Zhou, Y., Ram, P., Baracaldo, N., Samulowitz, H., Salonidis, T., and Chen, T. Enhancing in-context learning via linear probe calibration. In Proceedings of The 27th International Conference on Artificial Intelligence and Statistics, pp. 307–315, 2024.

2)	Zhixiong Han, Yaru Hao, Li Dong, Yutao Sun, and Furu Wei. Prototypical calibration for few-shot learning of language models. In Proc. of International Conference on Learning Representations, 2023.

3)	Zhongtao Jiang, Yuanzhe Zhang, Cao Liu, Jun Zhao, and Kang Liu. Generative calibration for in-context learning. In Houda Bouamor, Juan Pino, and Kalika Bali (eds.), Findings of EMNLP 2023.

4)	M. Shen, S. Das, K. Greenewald, P. Sattigeri, G. Wornell, and S. Ghosh. Thermometer: Towards universal calibration for large language models. ICML 2024.

These papers focus on calibration for LLMs and could also serve as valuable baselines. While the first three aim to enhance LLM performance, particularly in the in-context learning (ICL) setting, 1) Abbas et al. (2024) and 4) Shen et al. (2024) notably also tackle the issue of Expected Calibration Error (ECE) metric. Thus, I respectfully disagree with the authors’ statement in lines 358-359 that “While such methods can decrease ECE, it does not help with improving model classification performance,” as these papers address both aspects.

**Limited baselines**: The baselines are too limited, and the papers I mentioned earlier should be included as valid baselines.


**Limited evaluation**: The dataset selection here appears quite restricted compared to related studies. For example, Batch Calibration (Zhou et al., 2023) utilizes 13 classification datasets, while Generative Calibration (Jiang et al., 2023) uses 12. Using a wider range of datasets would lead to a more thorough assessment and strengthen the study.

**Unfair comparison**: If we use two additional input samples in the prompt (as mentioned in lines 415-416), the zero-shot effectively becomes 2-shot, while three-shot becomes five-shot. I know we are not providing labels for the 2 additional samples but still the comparison of 0-shot/3-shot/10-shot LF-ICL with 0-shot/3-shot/10-shot ICL is a bit unfair since adding additional samples provides additional context/information.

**Reproducibility**: The authors did not provide the code to reproduce the results, which raises concerns about reproducibility.

**Suggestions**:

Previous studies have shown that ICL is sensitive to the choice of demonstrations used in the few-shot setting. It would be good to report standard deviation across different choices of the test demonstrations in Table 1 and Table 2, as done by (Contextual Calibration, Zhao et al., 2021), (Linear Probe Calibration, Abbas et al., 2024), and (Prototypical calibration, Han et al., 2023) etc.

In Table 1, it would have been nice to see the results for the 3 shot-agg and 10 shot-agg as well.


**Suboptimal Writing**: The writing in the paper is quite suboptimal, containing numerous grammatical errors and typographical errors. Here are some suggestions for improvement:

Lines 027-228: Redundancy: ‘LLMs’ abbreviation defined again. It was already defined in the abstract.

Lines 094-097: Several grammatical issues in the sentence

Lines 135-136: grammar/typo: use ‘accuracies are’ or ‘accuracy is’. Moreover, "regardless whether" should be "regardless of whether". It is better to use 'LLM' abbreviation as defined earlier in the abstract.

Line 164: grammar/typo: use 'an LLM' instead of 'a LLM'.

Lines 195-197: typo/grammar: "dataset" should be pluralized as "datasets.". typo: add space in the beginning of the sentence (i.e. between '...datasets.' and 'A smaller...').

Lines 234-236: grammar/typo: "incapable for describe" should be "incapable of describing"

Lines 268-269: grammar/typo: "it has trouble" should be "they have trouble": The subject "LLMs" is plural, so it should be followed by "they" rather than "it."

Lines 270-272: grammar/typo: "without trained on them" should be "without being trained on them". typo/grammar: "asking LM" should be "asking the LLM".

Lines 289-291: suggestion for better readability: change "Notice that even we are using words like ‘comparing’" to "Note that even though we use words like ‘comparing’". suggestion for better readability: Change "and sample 3" to "or sample 3" to improve readability.

Line 296: typo/grammar: "is because of" → "is that, due to".

Line 298: The phrase "where one would expect" is confusing in context; replacing it with "one would expect" directly makes it clearer.
Line 302: suggestion: "bias to the samples" → "bias toward samples".

Line 303-304: suggestion: "degradation of inference performance on later input samples" → "degradation in inference performance for samples appearing later in the input".

Line 341 (Figure 5): typo/grammar: "across 5 dataset." should be pluralized as "across 5 datasets."

Lines 346-347: grammar/typo: "F1 score and accuracy is also improved" should be "F1 score and accuracy are also improved."

Line 348: grammar/typo: "Though" → "Although".

Lines 367-368: typo: "referencs" → "references".

Line 485: typo/grammar: "restricts" instead of "restricted".

**Questions:**

**Clarification regarding Figure 1 setting**: I am not able to understand the reason behind the difference in observations between Figures 1(a) and 1(b). Why do they behave differently? What distinguishes the experimental settings in each case—is it the use of different datasets, or some other factor? These are important questions, and an apples-to-apples comparison is essential for clarity.

**With-labels version**: How would the method perform if we included labels for the 2 additional input samples? Would adding labels increase/decrease the performance? If it improves, by how much? This would perhaps be an interesting study.

---

> ### Author Response · Authors · 2024-11-20
>
> We thank reviewer for carefully reading through our paper. We have addressed all the feedbacks as much as we can but there are some confusions that may lead to the misinterpretation of this paper's major contributions. It looks like review is mainly focusing on the 4.2 (post-hoc section) and our results (e.g. Table 2) but we'd like to point out the most highlighted parts are 4.1 and corresponding results table 1 and figure 1,2. Please check our comments below and the revised paper.
>
> For Weaknesses:
> 1. Key Related Works Missing: The statement (“While such methods can decrease ECE, it does not help with improving model classification performance,”) is only for the temperature scaling method (as claimed by the original paper, that it does not change classification accuracy).  Almost all other post-hoc calibration methods will have impacts on both accuracy and ECE. We emphasized the temperature scaling here, because the approach is very popular and easy to use (E.g. many LLM APIs provide temperature changes). To avoid the confusion, we made this statement clear in our revision. Now it’s: “While temperature scaling can decrease ECE, it does not help with improving model classification performance”
> 2. Limited baselines: This is where we believe the major confusion happened. We acknowledge that these papers are relevant to discuss but we don’t think it’s necessary to compare them in experiments as they are all post-hoc calibration only (not prompt level approach). We think there are some priorities that we want to restate. Our contributions are mainly three fold:
> i). identify indiscriminate miscalibration (Figure 1a) in LLMs
> ii). a prompt level label free method (main results in Table 1)
> iii). a post-hoc calibration method based on logits (main results in Table 2)
> i)&ii) with Figure 1 and Table 1 are the more novel&important findings. iii). is inspired/built on top of existing post-hoc calibration methods as we claimed on line 366-367.. We are including iii). as a section here mainly because ii). naturally brought a LF post-hoc method that is new to the field.
> For the 4 papers that reviewer mentioned, they are all post-hoc calibration, not prompt level based method.  Outputs of our method ii), in theory, can be further calibrated with any of these methods or ours (iii) for post-hoc calibration. However, only ICL itself is a prompt based method (comparable with ii). When combining them, the results are simply expected to be better (similar to combining ICL+ours, ours+agg). We now added these papers and make it clear at the end of section 4.2.
> 3. Limited evaluation: There are some capacity constraints. We listed this in the section 7.
> 4. Unfair comparison: We think we can agree that the cost rank is: 3X + 3Y>3X>0. We are not calling 2 input samples 2-shot because there’s no input-answer information provided and shouldn’t be referred to as few-shot. Few shot prompting is mostly referred to when answers are provided (e.g. https://arxiv.org/pdf/2301.00234).
> And more importantly, if you compare LF-10-shot, with regular 10-shot, then yes, additional unlabeled samples are the extra cost. But compared with labels, these are much easier to access. For example, you can use other training samples. On the other hand, LF-ICL 0-shot also outperforms or on-par with regular ICL 3-shot (e.g. F1 for Table 1 L2-2B, L3-8B), and LF-ICL 3-shot also outperforms or on-par regular 10-shot (e.g. F1 for Table 1 L3-8B, L3-70B) as well. For ECE, LF-ICL generally outperforms regular few-shot (with more demonstrations).
> 5. We addressed all the suboptimal writing in the revised paper.
>
> For Questions:
> 1. Figure 1 uses simulated data for illustrating the indiscriminate calibration and regular miscalibration with different probabilities patterns (similar probabilities across different samples in 1(a) and different probabilities across different samples in 1 (b)). Experiment setting uses the real world datasets with actual LLM inference.
> 2. For With-labels version:
> Can you clarify this question? For LF-0-shot, if adding labels for 2 additional input samples, that’s gonna be like LF-2-shot.

---

> ### Comment · Reviewer_H97s · 2024-11-22
>
> **1.	Key Related Works Missing and 2. Limited baselines**
>
> I have gone through the paper thoroughly and I do acknowledge the main contribution in Table 1 and Figures 1, 2, that includes identifying indiscriminate miscalibration, a label free method, and the pos-hoc method using logits. However, I am still not convinced that the authors did not use the mentioned papers as baselines. While these papers are post-hoc methods, they even compare with the raw prompt based ICL method. If authors claim that ‘When combining them, the results are simply expected to be better ‘, I expect to see these results.
>
> **4.Unfair comparison.**
>
> By including two additional input samples in the prompt, the zero-shot setup effectively becomes 2-shot, and the three-shot setup becomes five-shot. While it’s true that these additional samples are provided without labels, they still contribute additional context and information. I believe it is crucial to ensure an apples-to-apples comparison by using fairer baselines that incorporate the same amount of information, regardless of whether the information includes labels or not. This would make it clear whether the improvements come from the method itself or simply from the extra information.
>
> **Can you clarify this question? For LF-0-shot, if adding labels for 2 additional input samples, that’s gonna be like LF-2-shot.**
>
> Yes, I am referring to LF-2-shot. It would be nice to include such ablations to see the impact of adding label information.
>
>
> I appreciate that the authors have improved the writing per my feedback, so I’ve raised the Presentation score to 3.
> However, considering the aforementioned reservations, I am not entirely convinced and will respectfully maintain my rating.

---

### Official Review · Reviewer_kE25 · 2024-10-31

**Soundness:** 3
**Presentation:** 3
**Contribution:** 3
**Rating:** 6
**Confidence:** 5

**Summary:**

This paper addresses the problem of indiscriminate miscalibration in Large Language Models (LLMs). The authors introduce a new calibration metric called Discriminate KL Divergence (DKL) and propose an in-context comparative inference method -- a label-free approach termed LF -- to mitigate this issue. They validate their method on five classification tasks using five models from the LLama family, employing both traditional and their newly proposed metrics.

**Strengths:**

1. The idea of using comparative inference to improve confidence calibration in LLMs is novel.
2. Extensive experiments demonstrate the effectiveness of their method.
3. The paper is clearly written and easy to follow.

**Weaknesses:**

1. Lack of in-depth analysis on why comparative inference alleviates indiscriminate miscalibration. It remains unclear how such a comparison triggers the LLM to output discriminative confidence.
2. Unfair comparisons in the experiments. For instance, the 0-shot-agg LF-ICL requires ensembling 10 inferences to produce one result, resulting in 10 times the computational cost compared to the 0-shot ICL baselines. While this is not a major issue since 0-shot LF-ICL still outperforms 0-shot ICL, including a 0-shot-ensemble ICL baseline would provide a more rigorous comparison.

**Questions:**

1. Regarding the experiments on post-hoc calibration, the authors claim that "comparative inference with post-hoc calibration can be combined with few-shot prompting" to further improve performance. Do the authors have experimental data to support this claim?
2. The LF-based methods do not show a robust improvement in terms of Expected Calibration Error (ECE) in the post-hoc calibration experiments. Why do the authors not present results using other metrics like the DKL they have just proposed?

---

> ### Author Response · Authors · 2024-11-20
>
> We would like to thank the reviewer for their detailed evaluation of our manuscript and for providing valuable insights. We now added a new section for addressing the major concern of lacking in-depth analysis. Please check below for comments and our revised paper.
>
> For Weaknesses:
>  1. To better provide better explanations. We added a new section in the results (section 5,4). We used the LF-0 shot VS regular 0-shot as an ablation study. Our hypothesis is that by providing comparison samples, it enforces models to contrast and compare. With multiple variations and  differences of samples exposed, the model's confidence is ‘regularized’ and calibrated. To see this, we plot the bootstrapped (for balancing the correct and incorrect samples) histogram of probability of correct prediction and incorrect prediction . And we can see that with our method, the differences are more outstanding.
> As evidence that the model indeed is comparing and contrasting different samples, we provide a breakdown across different datasets. We found that for the financial dataset, such contrast is more obvious. This is likely because the financial task is an ‘ordinal’ dataset where it has ‘positive’ and ‘negative’ labels where ‘contrast’ and ‘comparison’ are supposed to be more effective in practice (in terms of showing the difference of probabilities, not accuracy). And we do observe that the probabilities are more outstanding there.
> 2. This is also brought up by another reviewer. We will make this clear in the paper. We listed ensemble (agg) methods; it's a natural method introduced without comparative inference. But that should not be the highlight of the paper.
>
> For Questions:
> 1.Yes, LF-0-shot is our method alone. LF-3-shot is our method combined with few-shot prompting. You can also check appendix 1 for a real example to see what we used for prompt and how we combined the method.
> 2. We presented DKL in the figure 3 with MACROCE (as these are new proposed and a relatively new metric in the field focused on the special patterns of miscalibration). ECE is mostly used in literature as a general metric measuring calibration so we listed it as a main metric in the table along with accuracy.

---

### Official Review · Reviewer_onKS · 2024-11-01

**Soundness:** 2
**Presentation:** 3
**Contribution:** 2
**Rating:** 3
**Confidence:** 4

**Summary:**

This paper investigates a phenomenon called "indiscriminate miscalibration" in LLMs during in-context learning, where models assign similar confidence levels to both correct and incorrect predictions. The authors propose new metrics to measure this issue and introduce a label-free comparative inference method that includes unlabeled samples in prompts to improve calibration and classification performance. The authors evaluate their approach on 5 datasets using various Llama models and demonstrate improvements in both classification metrics (F1 score, accuracy) and calibration metrics.

**Strengths:**

1. The paper identifies an interesting and important problem regarding LLM calibration
2. The proposed label-free approach is novel and doesn't require labeled examples
3. The method appears to improve both classification performance and calibration metrics

**Weaknesses:**

1. The experimental design has several limitations that raise concerns about the generalizability and robustness of the results. Firstly, the evaluation is only conducted on Llama family models, which may not be representative of other language models or architectures. Secondly, the 5 datasets used are relatively standard classification tasks, and it is unclear how the method would perform on more complex or challenging datasets. Furthermore, the maximum of 500 test samples feels limited for a robust evaluation, and the lack of ablation studies examining the impact of different comparison sample selection strategies makes it difficult to disentangle the effects of different components of the method.

2. The paper lacks a rigorous theoretical analysis of why the comparative inference approach works. The assumptions made in Section 4.1.2 about bias averaging (Equation 3) are not well justified, and it is unclear what underlying principles drive the improvement in calibration and classification performance. A more thorough exploration of the theoretical foundations of this method would be necessary to fully understand its implications and limitations.


3. The comparative inference method increases the inference cost significantly due to multiple forward passes, which may be a concern for real-world applications where computational resources are limited. Moreover, the post-hoc calibration requires validation data, which may not be readily available for open-ended generation tasks, limiting the method's applicability in these scenarios.

**Questions:**

1. Why do you believe the bias averaging assumption in Equation 3 holds? Can you provide theoretical or empirical evidence?

2. How do you select comparison samples in practice? Is there a strategy better than random selection?

3. What is the computational overhead of your method compared to standard few-shot prompting? Please provide detailed timing analysis.

4. How sensitive is the method to the choice of comparison samples?

5. Can you comment on the method's impact on open-ended text generation and language modeling tasks?

---

> ### Author Response · Authors · 2024-11-20
>
> We appreciate reviewer's constructive feedback. We think there's a major confusion that lead to reviewer think our method has more cost which we clarified here and we also address some comments in the new revision, please see our comments below and revised paper.
>
> For Weaknesses:
> 1. We wanted to evaluate the model on multiple model sizes and the Llama family is the best choice to our knowledge and it can provide a good comparison across the model sizes to draw conclusions (as less bias is introduced by different model families). Different models and more datasets are definitely worth trying, but we are limited by some capacity constraints. We acknowledge this is one of the limitations in section 7.
> 2. This is a good point. We don’t have strong justification that bias can be averaged out (e.g. to 0) but this is just an assumption. The theoretical analysis corresponding to this would be figure 5 where you can see the metrics move with more aggregation examples. Though it plateaus at certain points which requires further studies. We list this as future work in the discussion section.
> 3. This is an important point and we appreciate the reviewer for pointing this out. Please check 4.1.1, we always list samples to infer as the first example. You can cut the output after the first word as that’s the only answer you will need which is what we used for experiments as well. And if LLM is not generating more tokens/answers  (for comparison samples) and there will be no multiple forward passes and no additional cost. We made this clear in our paper now (end of section 4.1.1).
>
> For Questions:
> 1. We added a new section in the results (section 5,4) as an ablation section for understanding the behavior of our method and showed that Financial dataset is an ‘ordinal’  task that can be more sensitive/effective to the comparison approach (Please see the revision or the respond below for the second reviewer). Also,  figure 5 is the evidence for the aggregation part. We are inspired by some heuristic examples. For instance, considering a sentiment question, the outcome can be positive or negative. If you compare your sample with a very positive example, then LLM might be biased to give a negative answer and vice versa. But if you average them, then it’s averaged out. This would suggest that in aggregation, we need to draw samples from a balanced distribution for comparisons. In our studies, we simply random sampled from the data source we have.
> 2. This is a very interesting topic and we believe for each dataset, LLM, there may have very different conclusions. There should be a better strategy but that may also indicate additional cost. We listed this as future work (See section 7).
> 3. Copied from another response: This is an important point and we appreciate the reviewer for pointing this out. Please check 4.1.1, we always list samples to infer as the first example to avoid bias. You can cut the output when it generates the answer for your input. And if LLM is not generating more tokens (for comparison samples) and there will be no multiple forward passes and no additional cost.
> 4.similar to 2.
> 5. Are you asking for the next token/word prediction type of tasks? We believe this is interesting but not sure until someone really tries it. Challenges would come from: 1) need a proper prompt for comparison, can potentially do something like ‘comparing the following words as the next word’, etc. 2) Need to have a good pool of candidates for comparison. An heuristic is to first run inference with original text and prompt (without comparison) and then choose a few from the ones that have higher probabilities as possible ones for comparison. (Though low probability ones can also be useful but will require some handcraft work for selection). This is not very straightforward compared with well-defined supervised classification tasks though.

---

### Meta-Review · Area_Chair_6dJ8 · 2024-12-17

**Metareview:**

This paper investigates a phenomenon called "indiscriminate miscalibration" in LLMs during in-context learning, where models assign similar confidence levels to both correct and incorrect predictions. Motivated by this, the authors propose new metrics to measure this issue and introduce a label-free comparative inference method that includes unlabeled samples in prompts to improve calibration and classification performance. Two of the three reviewers raised concerns about limited experiments, lack of rigorous theoretical analysis, missing baselines and comparison on inference cost, and strongly recommended rejection. At the end of the rebuttal, the major concerns of the reviewers remained. Thus, this paper can not be accepted by ICLR in its current version.

**Additional Comments On Reviewer Discussion:**

Two of the three reviewers raised concerns about limited experiments, lack of rigorous theoretical analysis, missing baselines and comparison on inference cost, and strongly recommended rejection. At the end of the rebuttal, the major concerns of the reviewers remained. Two of the three reviewers strongly recommended rejection, putting this paper below the acceptance bar.

---

### Decision · Program_Chairs · 2025-01-22

Reject